# Evaluating the Effect of Intensity Standardisation on Longitudinal Whole Brain Atrophy Quantification in Brain Magnetic Resonance Imaging

**Emily E. Carvajal-Camelo** [1,*] , **Jose Bernal** [2,*] , **Arnau Oliver** [3,4] , **Xavier Lladó** [3,4] , **María Trujillo** [1] **and The Alzheimer's Disease Neuroimaging Initiative** [†]

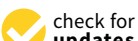

1 Multimedia and Computer Vision Group, Universidad del Valle, Cali 760001, Colombia; maria.trujillo@correounivalle.edu.co
2 Centre for Clinical Brain Sciences, The University of Edinburgh, Edinburgh EH16 4ST, UK
3 Computer Vision and Robotics Institute, Universitat de Girona, 17003 Girona, Spain; arnau.oliver@udg.edu (A.O.); xavier.llado@udg.edu (X.L.)
4 Red Española de Esclerosis Múltiple (REEM), 28040 Madrid, Spain
* Correspondence: emily.carvajal@correounivalle.edu.co (E.E.C.-C.); jose.bernal@ed.ac.uk (J.B.)
† Membership list can be found in the Acknowledgments section.

**Abstract:** Atrophy quantification is fundamental for understanding brain development and diagnosing and monitoring brain diseases. FSL-SIENA is a well-known fully automated method that has been widely used in brain magnetic resonance imaging studies. However, intensity variations arising during image acquisition may compromise evaluation, analysis and even diagnosis. In this work, we studied whether intensity standardisation could improve longitudinal atrophy quantification using FSL-SIENA. We evaluated the effect of six intensity standardisation methods—z-score, fuzzy c-means, Gaussian mixture model, kernel density estimation, histogram matching and WhiteStripe—on atrophy detected by FSL-SIENA. First, we evaluated scan–rescan repeatability using scans taken during the same session from OASIS ($n = 122$). Except for WhiteStripe, intensity standardisation did not compromise the scan–rescan repeatability of FSL-SIENA. Second, we compared the mean annual atrophy for Alzheimer's and control subjects from OASIS ($n = 122$) and ADNI ($n = 147$) yielded by FSL-SIENA with and without intensity standardisation, after adjusting for covariates. Our findings were threefold: First, the use of histogram matching was counterproductive, primarily as its assumption of equal tissue proportions does not necessarily hold in longitudinal studies. Second, standardising with z-score and WhiteStripe before registration affected the registration performance, thus leading to erroneous estimates. Third, z-score was the only method that consistently led to increased effect sizes compared to when omitted (no standardisation: 0.39 and 0.43 for OASIS and ADNI; z-score: 0.45 for both datasets). Overall, we found that incorporating z-score right after registration led to reduced inter-subject inter-scan intensity variability and benefited FSL-SIENA. Our work evinces the relevance of appropriate intensity standardisation in longitudinal cerebral atrophy assessments using FSL-SIENA.

**Keywords:** intensity standardisation; FSL-SIENA; longitudinal atrophy quantification; brain magnetic resonance imaging

## 1. Introduction

Longitudinal brain atrophy quantification is an active research area in medical image analysis as these measurements permit studying brain development, diagnosing brain diseases and evaluating disease progression over time, and assessing treatment effectiveness [1–6]. Although slow shrinkage of the brain comes with ageing, it may also be a neuroimaging feature of pathologies, such as schizophrenia, cerebral small vessel disease, Alzheimer's disease and multiple sclerosis [7–12]. Therefore, accurate and reliable brain

volume measurements are essential for characterising normal and abnormal brain tissue changes and understanding the nature of brain problems.

Longitudinal brain volumetry assessments consist of finding and quantifying brain tissue variations between two scans of the same patient taken at different time-points: a *baseline* and a *follow-up* scan, e.g., scans acquired at inclusion and a year later. Numerous algorithms have been proposed for carrying out such evaluations [8]. However, FSL-SIENA (Structural Image Evaluation, using Normalisation, of Atrophy) [13] continues to be a widespread tool in the medical community, given the fact that it is fully automated and available under open source license [8].

Like many other methods, FSL-SIENA computes brain edge displacement as a surrogate measure of atrophy [8]. The processing pipeline comprises skull stripping, registration, tissue segmentation and brain edge displacement estimation. Naturally, the accuracy and precision of each step determines the overall performance of FSL-SIENA. For example, intensity variations between baseline and follow-up scans—e.g., caused by imaging protocol—may have serious consequences on both the registration and segmentation steps, affecting subsequent stages of the analysis [14,15].

Previous works have explored ways to reduce intensity variations within FSL-SIENA [14,16]. Shah et al. [16] demonstrated the effectiveness of histogram matching for improving multiple sclerosis lesion segmentation in a multi-site multi-scanner setup. Battaglini et al. [14] showed the relevance of improved brain extraction and intensity correction modules. In both aforementioned works, the intensity standardisation step consisted of a piece-wise linear histogram matching, which assumes that the balance of tissue classes is consistent between subjects being matched. However, this assumption does not necessarily hold in longitudinal studies [17].

In this work, we evaluate the effect of intensity standardisation strategies on longitudinal atrophy quantification. In particular, we consider six strategies that are used in medical image analysis; all of them available to the public [17,18]. We hypothesise that incorporating intensity standardisation in atrophy quantification assessments leads to significantly better estimations compared to when omitted. To our best knowledge, this is the first time such an analysis has been carried out. The contributions of this work are threefold: (i) we showcase and make publicly available a ready-to-use tool for assessing the effect of intensity standardisation on scan–rescan and longitudinal atrophy quantification, (ii) we benchmark six intensity standardisation techniques for harmonising intensities between baseline and follow-up scans within a standard whole brain atrophy quantification pipeline and (iii) we show quantitatively that intensity standardisation may lead to improved longitudinal atrophy quantification.

## 2. Materials and Methods

### 2.1. Datasets

We considered two publicly available longitudinal MRI repositories: Open Access Series of Imaging Studies (OASIS) [19] and Alzheimer's Disease Neuroimaging Initiative (ADNI) [20]. For the sake of reproducibility, we attach the list of selected cases from ADNI and OASIS as Supplementary Materials.

OASIS. We used a subset of the OASIS2 longitudinal dataset comprising scans from 122 different subjects. In each imaging session, each subject was scanned three to four times using an MP-RAGE sequence and a 1.5T Vision scanner (Siemens, Erlangen, Germany). Imaging acquisition details can be found in Table 1. All subjects in the study were evaluated using the clinical dementia rating to determine their dementia status and classified into non-demented or very mild to mild Alzheimer's disease. We downloaded the complete OASIS2 longitudinal dataset and selected patients who remained in either category throughout the entire study and had their follow-up scan approximately between six months and three years of their baseline visit, as depicted in Figure 1. The resulting sample consisted of 60 Alzheimer's disease patients (mean age (SD) = 75.05 (6.89); female proportion = 47%) and 62 control subjects (mean age (SD) = 75.40 (8.70); female proportion = 74%).

ADNI. We used a subset of the ADNI longitudinal dataset comprising scans from 147 different subjects. Each subject was scanned using an MP-RAGE sequence and multiple scanners (General Electric (GE) Healthcare; Philips Medical Systems; and Siemens Medical Solutions). Imaging acquisition details can be found in Table 1. All subjects in the study were evaluated via global, functional and behavioural assessments, including the clinical dementia rating, to determine their dementia status. We used the advanced search tool in the ida.loni.usc.edu platform to extract our subsample. The inclusion criteria were image type = original (raw); field strength = 1.5T; slice thickness = 0.5–1.9 mm, weighting = T1, image description = MP-RAGE, subject group = AD or NC; visits = ADNI screening and ADNI1/GO Month 12. The resulting sample consisted of 64 Alzheimer's patients (mean age (SD) = 75.58 (8.16); female proportion = 45%) and 83 control subjects (mean age (SD) = 77.59 (4.56); female proportion = 53%). Subjects had their follow-up scan approximately one year after their baseline visit, as illustrated in Figure 1.

The ADNI was launched in 2003 as a public–private partnership, led by Principal Investigator Michael W. Weiner, MD. The primary goal of ADNI has been to test whether serial magnetic resonance imaging (MRI), positron emission tomography (PET), other biological markers, and clinical and neuropsychological assessment can be combined to measure the progression of mild cognitive impairment (MCI) and early Alzheimer's disease (AD). For up-to-date information, see www.adni-info.org [accessed on 10 February 2021]).

**Table 1.** Image acquisition details of considered OASIS and ADNI scans. We extracted the information in this table from the publications in [19,20].

| Parameter | OASIS | ADNI |
|---|---|---|
| Sequence | MP-RAGE | MP-RAGE |
| Repetition time (ms) | 9.7 | 3000 |
| Echo time (ms) | 4.0 | – |
| Flip angle | 10° | 8° |
| Inversion time (ms) | 20 | 1000 |
| Orientation | Sagittal | Sagittal |
| Thickness (mm) | 1.25 | 1.20 |
| Slice number | 128 | 184–208 |
| Resolution | 256 × 256<br>1×1 mm | 192 × 192<br>1.25 × 1.25 mm |

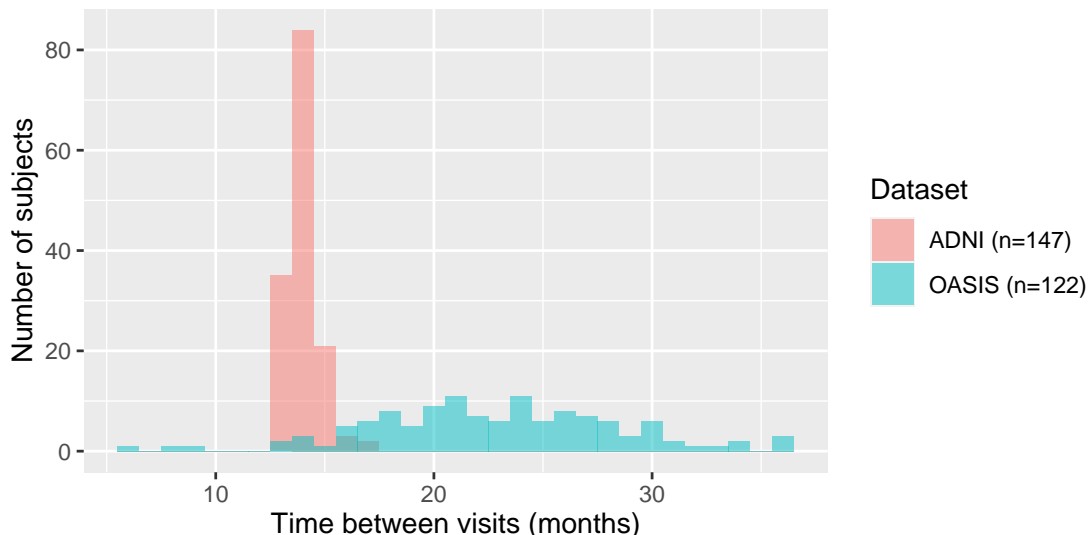

**Figure 1.** Distribution of time between visits in the considered ADNI and OASIS datasets.

### 2.2. Equipping FSL-SIENA with Intensity Standardisation

We quantified atrophy using FSL-SIENA [13]. This tool measures brain edge displacement between scans acquired at baseline and follow-up visits as a surrogate measure of cerebral atrophy. Briefly, the processing pipeline consists of skull stripping, registration, tissue segmentation and brain edge displacement analysis. Among these steps, intensity variations have been found to affect registration [21] and segmentation [14]. Thus, in this work, we evaluate the effect of standardising intensities in FSL-SIENA in two separate experiments: when standardisation occurs before registration (Figure 2a) and after registration (Figure 2b). Given that FSL-BET (default skull stripping tool) may not perform well in Alzheimer's disease patients [22], we used ROBEX instead due to its improved effectiveness [23,24].

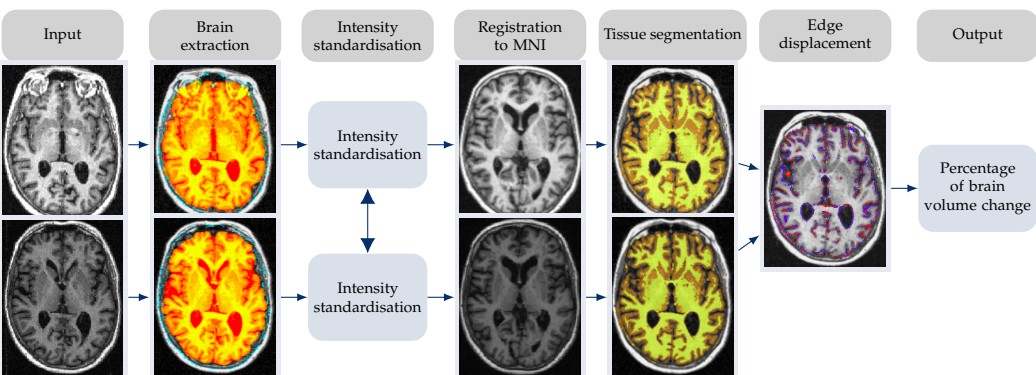

(**a**) Intensity standardisation before registration

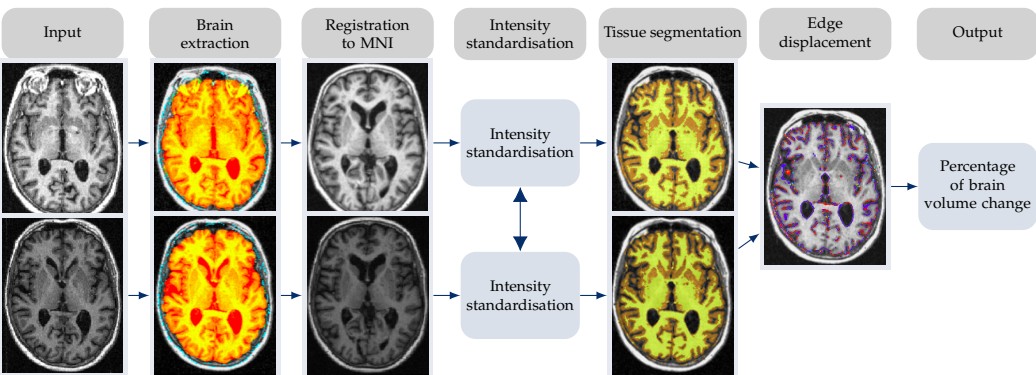

(**b**) Intensity standardisation after registration

**Figure 2.** Incorporating intensity standardisation in the FSL-SIENA pipeline. Standardisation takes place before or after registering input volumes to the MNI space. The inputs correspond to the baseline and follow-up T1-w scans. The FSL-SIENA pipeline consists of skull stripping, spatial normalisation, tissue segmentation, edge displacement analysis and brain volume change reporting.

### 2.3. Considered Intensity Standardisation Techniques
#### 2.3.1. z-Score

The z-score is a popular normalisation method that linearly transforms intensities to have zero mean and unit standard deviation. Given an input image $I$, the process consists of subtracting the mean intensity value from each voxel and dividing the result by the standard deviation value as follows,

$$I_{z-score} = \frac{I - \mu}{\sigma}, \tag{1}$$

where $\mu$ and $\sigma$ are the mean and standard deviation of intensities within the intracranial volume, respectively.

### 2.3.2. Fuzzy c-Means-Based Standardisation

The fuzzy c-means-based standardisation leverages fuzzy clustering of the white matter. The process is as follows. First, fuzzy c-means clustering takes place [25]. Given a set of voxels $x_i \in I$, fuzzy c-means finds a set of $K$ clusters, $C = \{c_1, ..., c_K\}$, minimising the following loss function:

$$\arg\min_{C} \sum_{x_i \in I} \sum_{c_k \in C} w_{ik} \cdot ||x_i - c_k||_2^2, \tag{2}$$

where $c_k$ is the centroid of the cluster $k$ and $w_{ik} \in [0, 1]$ is the membership of $x_i$ to the cluster $k$. The membership of a data point to a certain cluster is given by

$$w_{ik} = \frac{1}{\sum_{j=1}^{K} \frac{||x_i - c_k||_2^2}{||x_i - c_j||_2^2}}. \tag{3}$$

The process is carried out iteratively until convergence. Second, the method divides all voxel intensities by the mean intensity of the detected white matter [18]. The selection of $K$ may depend on the regions of interest to be segmented or on the presence of abnormal tissue (e.g., white matter hyperintensities, stroke lesions or tumours) or extra-cerebral regions. As in the original implementation, we set $K = 3$ (ideally white matter, grey matter and cerebrospinal fluid).

### 2.3.3. Gaussian Mixture Model-Based Standardisation

The Gaussian mixture model-based standardisation leverages supervised clustering to segment the white matter. This clustering technique assumes that the histogram of intensities can be represented using a mixture of Gaussian distributions for each region of interest. Intensities belong to a cluster depending on the number of standard deviations from the intensity value to the mean. Given a set of voxels $x_i \in I$, clustering consists of finding $K$ Gaussian distributions maximising the following loss function:

$$\arg\max_{(\pi, \mu, \sigma)} \sum_{x_i \in I} \ln \sum_{k=1}^{K} \pi_j \cdot \mathcal{N}(x_i \mid \mu_k, \sigma_k), \tag{4}$$

where the $k$-th Gaussian distribution of the mixture is represented by its mean $\mu_k$ and standard deviation $\sigma_k$ and has a weight $\pi_k$ in the mixture. Note that $\sum_{j=1}^{K} \pi_j = 1$. Similarly to the fuzzy c-mean standardisation, the method divides all voxel intensities by the mean intensity of the segmented white matter [18].

### 2.3.4. WhiteStripe

The WhiteStripe method proposed by Shinobara et al. [17] estimates statistics of intensities of the white matter to normalise all other regions of interest. The process is as follows. First, the method uses a penalised spline smoother to estimate the latest non-background mode of the histogram of intensities, $\mu_{WS}$. In T1-w scans, this mode would coincide with the white matter mode. Second, the method estimates the standard deviation, $\sigma_{WS}$ from a window of 10% of the intensity values around the detected mode. Third, all intensities are normalised in a z-score fashion using these two statistics (Equation (1)). This standardisation method has been shown to be robust to the presence of white matter lesions in ageing populations [17].

### 2.3.5. Kernel Density Estimation Based Standardisation

This method estimates a probability density function out of a set of data points (in this case, histogram of intensities). The probability density function, $p(x)$, is expressed as

$$p(x) = \frac{1}{card(I)} \sum_{x_i \in I} \mathcal{K}\left(\frac{x - x_i}{h}\right), \tag{5}$$

where $\mathcal{K}$ is the smoothing kernel, $h$ is the smoothing parameter and $card(\cdot)$ denotes cardinality. The contribution of each data point depends on its neighbourhood: the closer the points to $x$, the higher the $p(x)$. The bandwidth controls the smoothness of $p(x)$: the lower the $h$, the smoother the resulting probability density function. Once the method computes $p(x)$, it finds its right-most mode and assumes it represents the white matter mode. In this work, we used a Gaussian kernel and $h = \max(I)/80$ as provided in the implementation in [18].

### 2.3.6. Piecewise Linear Histogram Matching

The piecewise linear histogram matching method proposed by Nyul and Udupa [26] learns an intensity mapping function from a reference scan and uses it to standardise input scans. In the training step, the method identifies a set of $p$ landmarks (e.g., percentiles or modes) from the histogram of the reference scan and creates a piecewise linear function by interpolating linear segments between consecutive landmarks. The method avoids the minimum and maximum values to add robustness against outliers. In the transformation phase, the same landmarks are identified and intensities updated according to the estimated mapping function. In this work, we assumed the reference scan was the baseline scan of each subject.

### *2.4. Evaluation Analysis and Measures*

### 2.4.1. Quality of Intensity Standardisation

The Kullback–Leibler (KL) divergence measures the difference between two probability distributions, expressed as intensity histograms in this work. As the standardisation process seeks to map intensity distributions to a similar range, we use this metric as a measure of the standardisation quality. Given two probability density functions, $p$ and $q$, the KL divergence is computed as follows,

$$KL(p,q) = \sum_{i=1}^{N} p(x_i) \log \frac{p(x_i)}{q(x_i)}. \tag{6}$$

The more similar the distributions, the lower the KL divergence value. We compare the KL divergence between baseline and follow-up scans before and after standardisation using the Wilcoxon signed-rank test. We expect the KL divergence value to decrease after standardising intensities.

### 2.4.2. Scan–Rescan Repeatability

Scan–rescan assessments measure the robustness of an atrophy quantification algorithm against subtle imaging variations. As the atrophy level between scans of the same patient taken on the same visit with the same imaging protocol and same scanner should be minimal, the expect atrophy measured by FSL-SIENA should be close to zero. We study whether standardising intensities could affect the performance of the original method. We compare the scan–rescan error with and without equipping FSL-SIENA with intensity standardisation using the Wilcoxon signed-rank test.

### 2.4.3. Testing for Atrophy Differences between Alzheimer's Disease and Normal Control Subjects

The lack of manual segmentation limits assessing the accuracy of atrophy quantification methods [27]. The evaluation typically consists of testing for differences in atrophy rates between a pathological and a control group [22]. The evaluation process is as follows. We run FSL-SIENA on baseline and follow-up scans from OASIS and ADNI using FSL-SIENA with and without intensity standardisation. Next, we conducted analyses of covariance to determine a statistically significant difference between Alzheimer's disease and normal control subjects on detected atrophy rates. We controlled for age, biological sex, time between visits and normalised baseline brain volume (total brain volume divided by intracranial volume). Then, we estimated the marginal mean brain volume change at

one year of inclusion using the resulting linear models. Finally, we estimated effect size using the Cohen's *d* formula:

$$\text{Cohen's d} = \frac{\mu_{AD} - \mu_{NC}}{\sqrt{(\sigma_{AD}^2 + \sigma_{NC}^2)/2}}, \tag{7}$$

where $\mu_{AD}$ and $\mu_{NC}$ are the mean atrophy rates for the Alzheimer's disease and the normal control groups, and $\sigma_{AD}$ and $\sigma_{NC}$ are their standard deviations. Given that the sample under consideration is fixed, an increase in effect size after standardisation could imply that either inter-group differences or intra-group homogeneity has increased. In practice, this situation would translate into smaller sample sizes to show statistical differences between Alzheimer's disease and control groups.

### 2.5. Implementation Details

We implemented all methods in Python using the Intensity Normalisation library [18] and used FSL-SIENA v6.0. We ran all the experiments on a GNU/Linux machine box running Ubuntu 18.04, with 16 GB RAM. The developed framework is available to download at our Github repository (https://github.com/emyesme/IntensityStandarisation [accessed on 10 February 2021]). We carried out all statistical analyses using RStudio v1.1.456 with R v3.5.1.

## 3. Results

We ran FSL-SIENA on both ADNI and OASIS before and after standardisation with the proposed pipelines shown in Figure 2 and assessed the effect of the additional processing step on scan-rescan repeatability and prediction of dementia status. Furthermore, we evaluated the similarity between histogram of intensities of baseline and follow-up scans before and after standardisation using the KL divergence as a surrogate measure of the standardisation quality. The experimental results are described in the following sections.

### 3.1. Quality of Intensity Standardisation

We measured the quality of standardisation by determining whether histograms of baseline and follow-up scans were similar, in terms of the KL divergence, after standardising intensities or not. The measured KL distance is shown in Figure 3. In most cases, standardising intensities between baseline and follow-up scans resulted in significantly lower KL divergence values ($p < 0.001$), except when using WhiteStripe. The situation may be explained by the large variation in the left-most tails of the resulting histograms of intensities (See Appendix A). This variation results in a higher mismatch between histograms of intensities and, thus, in higher KL divergence.

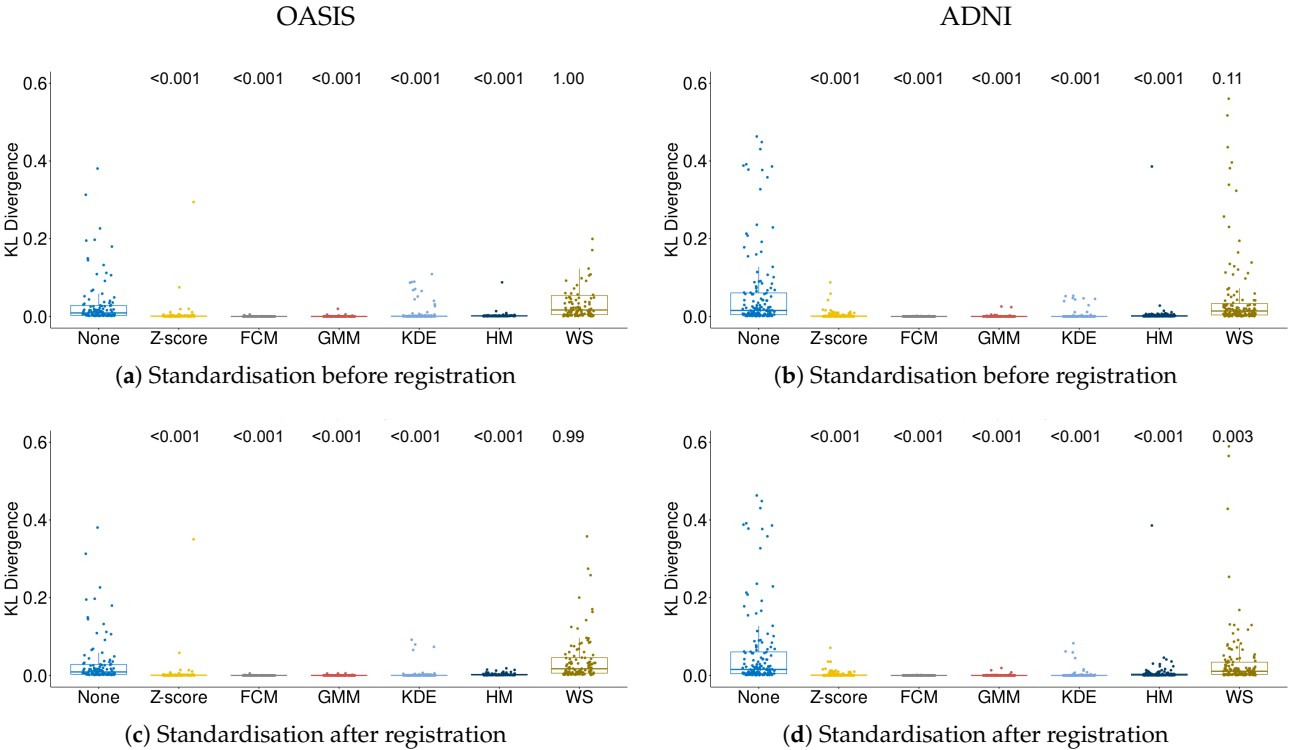

**Figure 3.** Similarity between histograms of intensities with and without standardisation. We equipped FSL-SIENA with intensity standardisation before or after registration. We measured the degree of similarity between the histogram of intensities of baseline and follow-up using the Kullback–Leibler (KL) divergence. We used the Wilcoxon signed-rank test to examine whether the median KL divergence obtained after standardisation was lower than that obtained when omitted (*p*-values on top). FCM: Fuzzy c-means. GMM: Gaussian mixture model. KDE: Kernel density estimation. HM: Histogram matching. WS: WhiteStripe.

### 3.2. Scan–Rescan Repeatability

The scan–rescan repeatability experiment consisted of examining whether intensity standardisation could result in increase scan–rescan error compared to when omitted. We ran FSL-SIENA on pairs of scans of subjects from OASIS, that were acquired in a single session (as explained in Section 2.1). We used the resulting percentage of brain volume change yielded by FSL-SIENA as measure of error as this variation should be close to zero. The results are presented in Figure 4. While most intensity standardisation methods did not compromise the original robustness of FSL-SIENA against subtle imaging artefacts, WhiteStripe did ($p < 0.001$). Furthermore, the application of intensity standardisation before registration led to less dispersion of scan-rescan errors overall.

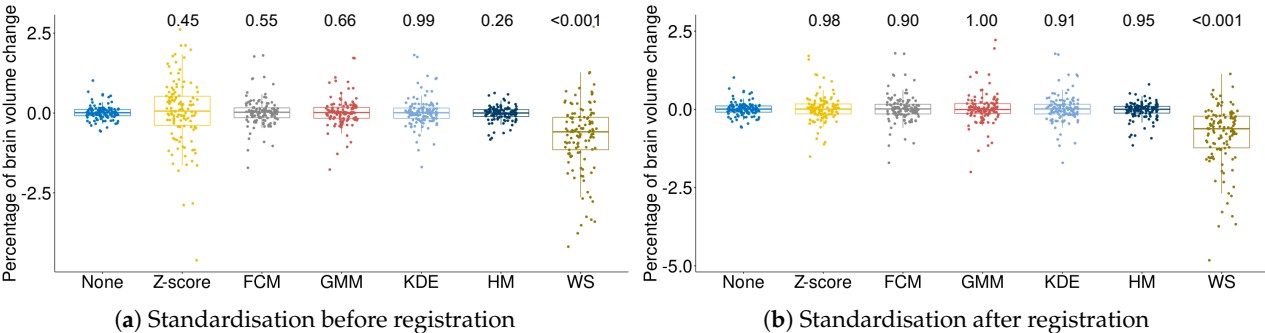

**Figure 4.** Scan–rescan deviation with and without intensity standardisation. We equipped FSL-SIENA with intensity standardisation before or after registration. We expressed the error as the percentage of brain volume change between scans acquired during the same visit and using the same scanner and acquisition protocol. We used the Wilcoxon signed-rank test to examine whether the median error obtained after standardisation differed from that obtained without standardisation (*p*-values on top). FCM: Fuzzy c-means. GMM: Gaussian mixture model. KDE: Kernel density estimation. HM: Histogram matching. WS: WhiteStripe.

*3.3. Effect of Intensity Standardisation on Atrophy Differences between Alzheimer's Disease and Normal Control Subjects*

We evaluated the effect of intensity standardisation on atrophy differences between the pathological and control groups in ADNI and OASIS. The procedure was as follows. First, we ran FSL-SIENA, with and without intensity standardisation, on pairs of baseline and follow-up scans. We tested the effect of standardising either before or after registration, as illustrated in Figure 2. Second, we tested for differences in atrophy between the two groups, after adjusting for age, biological sex, baseline normalised brain volume (total brain volume divided by intracranial volume) and time between visits. We report the estimated mean annual atrophy for both groups and whether differences between estimates were significant or not in Table 2.

The estimated annual atrophy for Alzheimer's disease patients was higher than that of the control group regardless of the method and dataset. Nevertheless, the values themselves varied depending on the dataset under examination. We observed that atrophy gauged in OASIS disagreed with that in ADNI. Brain tissue loss measured in subjects in the former dataset was consistently lower than that gauged in ADNI, regardless of whether standardisation was considered or not. For example, in the absence of standardisation, the mean annual brain volume change for Alzheimer's disease and control subjects in OASIS was −0.61 (95% CI −0.85, −0.37) % and −0.10 (95% −0.35, 0.17) %, while estimates for the same groups were equal to −0.89 (95% CI −1.10, −0.69) and −0.27 (95% −0.45, −0.08) in ADNI. Further inspection of other clinical parameters revealed that the mean normalised baseline brain volume differed between datasets: mean values were approximately 0.58 (95% CI 0.57, 0.59) % and 0.55 (95% CI 0.55, 0.56) % for the pathological and control group in ADNI and 0.72 (95% CI 0.71, 0.73) % and 0.75 (95% CI 0.74, 0.76) % in OASIS. Additionally, image acquisition parameters (Table 1) could have contributed to these inter-dataset atrophy variations as these parameters determine tissue contrast and partial volume effects, which could have led to a distinct characterisation of brain boundaries in each dataset.

Estimated annual brain volume changes also differed depending on whether intensity standardisation was carried out or not. We noticed two clear trends in this regard. First, the application of histogram matching resulted in more similar estimates for both Alzheimer's disease patients and normal control subjects, even more similar than those detected without standardisation. This situation may be reflective of a potential under-detection of the actual atrophy due to information loss (e.g., contrast loss) after the application of histogram matching. Second, standardising intensities with z-score and WhiteStripe before registering baseline and follow-up scans resulted in lower effect sizes compared

to those obtained when applying them afterwards (z-score: 0.39 vs. 0.45 in OASIS and 0.27 vs. 0.45 in ADNI; WhiteStripe: 0.37 vs. 0.44 in OASIS and 0.02 vs. 0.46 in ADNI). After inspecting the registration outputs, we noticed that standardising with these two techniques prior to registration led to unsatisfactory outcomes in numerous cases (ADNI: 63 with WhiteStripe and 26 with z-score; OASIS: 24 with WhiteStripe and 44 with z-score). Some examples of this issue are shown in Figure 5. On the other hand, the application of these two techniques after registration led to some of the highest Cohen's *d* values in both datasets.

WhiteStripe before registration             z-score before registration

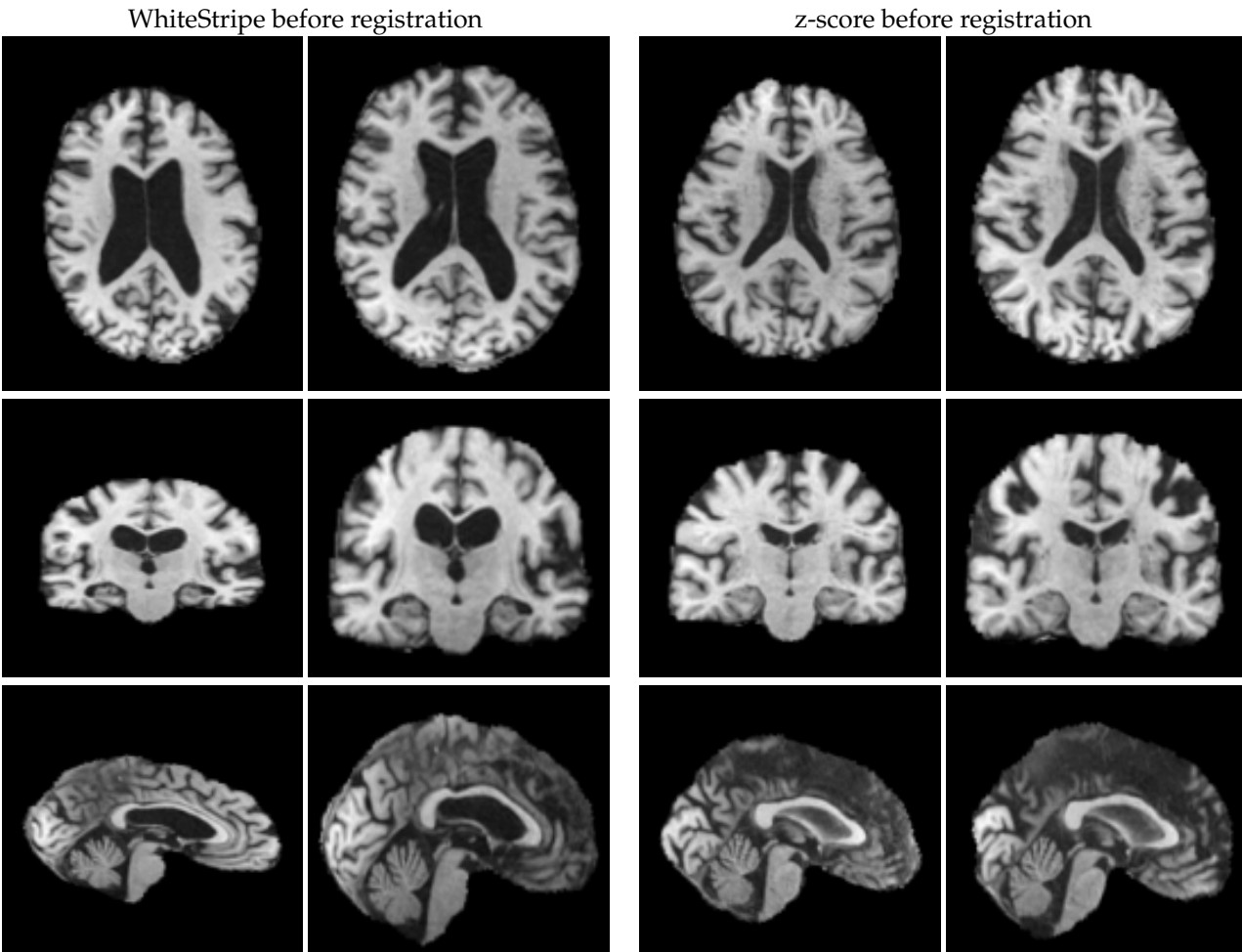

**Figure 5.** Examples in which registration failed due to the application of WhiteStripe and z-score before registration. The cases correspond to "ADNI_005_S_0814" and "ADNI_098_S_0149" for which brain volume change was estimated around 2.89% and −5.02%, respectively.

**Table 2.** Mean annual brain volume change (%) in Alzheimer's disease (AD) and normal control (NC) subjects estimated using FSL-SIENA with and without intensity standardisation. We obtained mean values after controlling for age, biological sex, baseline normalised brain volume (total brain volume divided by intracranial volume) and time between visits. Additionally, we tested whether mean atrophy for both groups was similar. We reported the effect size (Cohen's *d*) computed using Equation (7)) and *p*-value. CI: 95% confidence interval.

| | | | None | z-Score | Fuzzy c-Means | Gaussian Mixture Model | Kernel Density Estimation | WhiteStripe | Histogram Matching |
|---|---|---|---|---|---|---|---|---|---|
| OASIS (n = 122) | Before | Mean (CI), AD % | −0.61 (−0.85, −0.37) | −0.94 (−1.65, −0.23) | −1.00 (−1.51, −0.49) | −0.99 (−1.49, −0.49) | −0.98 (−1.49, −0.48) | −0.94 (−0.59, −1.28) | −0.59 (−0.79, −0.40) |
| | | Mean (CI), NC % | −0.10 (−0.35, 0.17) | 0.48 (−0.30, 1.26) | 0.06 (−0.50, 0.62) | 0.04 (−0.50, 0.60) | 0.10 (−0.46, 0.67) | −0.01 (0.38, −0.41) | −0.27 (−0.49, −0.06) |
| | | Cohen's d | 0.39 | 0.39 | 0.44 | 0.46 | 0.45 | 0.37 | 0.30 |
| | | *p*-value | 0.03 | 0.03 | 0.02 | 0.01 | 0.01 | 0.04 | 0.10 |
| | After | Mean (CI), AD % | −0.61 (−0.85, −0.38) | −1.01 (−1.49, −0.53) | −1.01 (−1.51, −0.50) | −1.05 (−1.62, −0.49) | −1.01 (−1.52, −0.50) | −0.83 (−1.18, −0.49) | −0.60 (−0.81, −0.39) |
| | | Mean (CI), NC % | −0.09 (−0.35, 0.17) | 0.03 (−0.49, 0.56) | 0.03 (−0.53, 0.59) | 0.08 (−0.54, 0.71) | 0.02 (−0.53, 0.59) | −0.04 (−0.42, 0.34) | −0.35 (−0.58, −0.11) |
| | | Cohen's d | 0.39 | 0.45 | 0.44 | 0.45 | 0.44 | 0.44 | 0.23 |
| | | *p*-value | 0.03 | 0.01 | 0.02 | 0.01 | 0.02 | 0.02 | 0.20 |
| ADNI (n = 147) | Before | Mean (CI), AD % | −0.89 (−1.10, −0.69) | −2.42 (−3.78, −1.07) | −1.43 (−1.76, −1.09) | −1.42 (−1.75, −1.09) | −1.44 (−1.77, −1.10) | −1.03 (−2.29, 0.21) | −0.77 (−0.95, −0.59) |
| | | Mean (CI), NC % | −0.27 (−0.45, −0.08) | 0.19 (−1.03, 1.42) | −0.44 (−0.75, −0.14) | −0.44 (−0.74, −0.14) | −0.45 (−0.75, −0.14) | −0.78 (−1.92, 0.36) | −0.28 (−0.44, 0.12) |
| | | Cohen's d | 0.43 | 0.27 | 0.42 | 0.42 | 0.42 | 0.02 | 0.39 |
| | | *p*-value | 0.02 | 0.11 | 0.02 | 0.02 | 0.02 | 0.89 | 0.02 |
| | After | Mean (CI), AD % | −0.89 (−1.10, −0.69) | −1.47 (−1.79, −1.14) | −1.45 (−1.78, −1.12) | −1.51 (−1.87, −1.14) | −1.45 (−1.78, −1.12) | −1.19 (−1.44, −0.93) | −0.71 (−0.89, −0.55) |
| | | Mean (CI), NC % | −0.27 (−0.45, −0.08) | −0.45 (−0.75, −0.15) | −0.45 (−0.75, −0.15) | −0.46 (−0.80, −0.12) | −0.44 (−0.75, −0.14) | −0.36 (−0.59, −0.13) | −0.27 (−0.41, −0.10) |
| | | Cohen's d | 0.43 | 0.45 | 0.45 | 0.41 | 0.42 | 0.46 | 0.36 |
| | | *p*-value | 0.01 | 0.003 | 0.008 | 0.02 | 0.01 | 0.007 | 0.03 |

## 4. Discussion

Recent works have shown that intensity non-standardness contributes to atrophy quantification errors [14]. In this work, we studied whether intensity standardisation could benefit an established and thoroughly validated atrophy quantification tool provided in the FSL package, FSL-SIENA. Given that both registration and segmentation may be affected by intensity variations, we tested the effect of incorporating intensity standardisation before registration and segmentation and before segmentation only. We considered six intensity standardisation techniques comprising z-score, fuzzy c-means, Gaussian mixture model, kernel density estimation, WhiteStripe and histogram matching. To our knowledge, this is the first time that the effect and suitability of multiple intensity standardisation has been quantitatively investigated for cerebral atrophy quantification.

We examined the degree of similarity between histograms of intensity of baseline and follow-up scans before and after standardisation as a measure of the quality of the standardisation process per se. Given that an intensity standardisation method maps histograms to a normalised space, we expected its use to lead to lower KL divergence

values. Indeed, we observed that inter-subject inter-scan variability was reduced with the use of most intensity standardisation methods, except for WhiteStripe for which the KL divergence was comparable to that measure without standardisation. A closer examination of the histograms of intensity in Figures A1 and A2 revealed that standardising intensities with WhiteStripe led to high coincidence in the white matter mode, but less coincidence in grey matter and cerebrospinal fluid modes. A similar observation was highlighted by Fortin et al. [28].

We assessed the effect of standardisation on the scan–rescan repeatability of FSL-SIENA by examining atrophy detected in scans of the same subject taken during the same imaging session. Given that the level of atrophy is expected to be minimal during short periods of time, a large deviation from zero would suggest the quantification method—not the standardisation method—is sensible to subtle imaging variations. Thus, we tested whether scan–rescan errors were similar before and after standardising intensities. Most intensity standardisation methods lead to median scan–rescan errors similar to those measured on raw scans, except for WhiteStripe ($p < 0.001$). Furthermore, standardising before registration and segmentation resulted in increased variance in scan–rescan errors, especially for z-score and WhiteStripe. Even though we did not detect a clear misregistration problem, we noticed subtle misalignment in a few cases that may have resulted in the measured errors.

We evaluated the effect of intensity standardisation on atrophy differences—not on histograms of intensity—gauged by FSL-SIENA between Alzheimer's disease and normal control subjects from ADNI and OASIS. A larger effect size implies that smaller sample sizes are required to show statistical significance between groups. We opted for this approximation as assessing accuracy in longitudinal atrophy quantification is difficult [27,29] due to the lack of manual segmentations and histological confirmation. According to our experimental results, intensity standardisation affected the estimation of brain atrophy. The atrophy gauged on standardised scans was larger than that measured on raw scans. Whether one or the other is more reflective of the disease cannot be clarified in this study. However, it is important to remark that mean annual brain volume changes detected on images standardised with multiple methods were similar. Furthermore, the use of z-score and WhiteStripe led to consistently higher effect sizes compared to those obtained without standardisation if applied after registration and before segmentation. We noticed that their use prior to registration led to a clear misregistration in numerous cases. As both z-score and WhiteStripe showed a similar behaviour and both standardisation strategies led to negative values, we hypothesise that negative values may have affected FSL-FLIRT's performance. Further research in this regard is needed to determine the specific step affected by this standardisation scheme.

z-score is one of the simplest intensity standardisation methods considered in this study. Despite being based on statistics which are sensitive to outliers, this strategy reduced inter-subject variations in both datasets, as depicted in Figure 3. If applied after registration, z-score could be included within FSL-SIENA.

Clustering-based intensity standardisation methods—fuzzy c-means and Gaussian mixture model—segment regions of interest in the brain and use the mean intensity of the selected region to standardise intensities. Leveraging on these types of techniques seemed compatible with FSL-SIENA as they reduced inter-scan intensity variability. However, low tissue contrast as well as the presence of brain abnormalities and extra-cerebral regions can weaken their performance. Therefore, we recommend visual inspection of input scans and histogram of intensities to determine whether clustering is feasible and the optimal number of clusters to reach optimal performance.

The histogram matching method proposed by Nyul et al. [26] assumes equal tissue proportions between the input scans [17], i.e., histograms of both the reference and input scan are equivalent. However, such an assumption does not necessarily hold in longitudinal studies due to the appearance of new brain lesions and brain tissue loss between visits.

Moreover, linear interpolation between landmarks results in potential information and contrast loss. Thus, we do not recommend its use in longitudinal studies using FSL-SIENA.

WhiteStripe standardises intensities based on the right-most mode (in T1-w scans) and, thus, ensures histograms of intensities will coincide at least on the detected peak. Furthermore, this methods does not require segmentation, thus reducing computation time and dependency on the accurate identification of the region of interest. Despite its theoretical advantages and improved performance in other scenarios [17], our experimental results suggest WhiteStripe is not compatible with FSL-SIENA as it can compromise its performance. Note that this outcome does not imply WhiteStripe is not a recommended intensity standardisation strategy, but that its use within FSL-SIENA compromises FSL-SIENA's performance. Additionally, we suggest using the original implementation of WhiteStripe in R as third-party implementations may not perform the same processing steps.

Kernel density estimation also processes the histogram of intensities to find the right-most mode (in T1-w scans). Unlike WhiteStripe, it normalises intensities by the maximum peak, i.e., its application results in positive values only. This might explain why the method did not affect FSL-FLIRT's performance. At an experimental level, we found that the incorporation of this intensity standardisation method to FSL-SIENA did not seem to compromise its scan–rescan performance. Furthermore, its application resulted in higher effect sizes, i.e., an increase differentiation between Alzheimer's disease and normal control patients. However, the performance of kernel density estimation depends on the smoothing parameter $h$: small values of $h$ may lead to spiky estimates while large values may misrepresent the multimodal nature of the data.

Our work has limitations. First, we assumed that higher atrophy differences between Alzheimer's disease and normal control subjects implied reduced errors in atrophy quantification compared to when omitted. Even though we found out pathological groups had higher atrophy rates than normal control groups with or without standardisation, in line with the literature [7], the values themselves varied. Whether one or the other is more accurate or reflective of the actual pathology is unclear and out of the scope of this work. In this regard, we suggest using atrophy generator pipelines in the future to have a sense of ground truth [30–32]. Second, we only used one method for assessing brain volume change, and thus it is difficult to generalise our findings to other medical image analysis tools. Furthermore, even though FSL-SIENA is highly accessible and relatively easy and ready to use, we are aware that it is not in the state-of-the-art of longitudinal atrophy quantification in multiple sclerosis and Alzheimer's disease, but the Jacobian determinant integration method [33,34]. Thus, future work should consider testing the pertinence of intensity standardisation on other automatic image analysis pipelines. Third, we did not analyse the effect of other confounding factors, such as intra-subject intensity standardisation methods (e.g., bias field correction methods). However, those are likely to have an effect on atrophy quantification. Thus, future work should analyse the effect of these types of algorithms on FSL-SIENA.

In conclusion, intensity standardisation can improve longitudinal whole-brain atrophy quantification using FSL-SIENA, but not all methods do. Their applicability depends to a great extent on whether their theoretical assumptions are met or not. We recommend incorporating z-score into FSL-SIENA, right after registration, to reduce inter-scan intensity variations as it is computationally efficient and improves FSL-SIENA's performance consistently.

**Supplementary Materials:** The following are available online at https://www.mdpi.com/2076-3417/11/4/1773/s1.

**Author Contributions:** Conceptualisation, E.E.C.-C. and J.B.; methodology, E.E.C.-C. and J.B.; software, E.E.C.-C. and J.B.; validation, J.B.; formal analysis, E.E.C.-C. and J.B.; investigation, E.E.C.-C. and J.B.; resources, M.T.; data curation, E.E.C.-C.; writing—original draft preparation, E.E.C.-C. and J.B.; writing—review and editing, E.E.C.-C., J.B., A.O., X.L., and M.T.; visualisation, E.E.C.-C., J.B., A.O., and X.L.; supervision, J.B. and M.T.; project administration, J.B.; funding acquisition, M.T. All authors have read and agreed to the published version of the manuscript.

**Funding:** This research was funded by the MRC Doctoral Training Programme in Precision Medicine (JB—Award Reference No. 2096671). This work has been partially supported by DPI2017-86696-R from the Ministerio de Ciencia, Innovación y Universidades.

**Institutional Review Board Statement:** Not applicable.

**Informed Consent Statement:** Not applicable.

**Data Availability Statement:** We have made our source code publicly available at https://github.com/emyesme/IntensityStandarisation [accessed on 10 February 2021]. Also, we have attached the list of selected cases from ADNI and OASIS as supporting documents to enable reproducibility.

**Acknowledgments:** Data used in the preparation of this article were [in part] obtained from the OASIS dataset: Longitudinal: Principal Investigators: D. Marcus, R, Buckner, J. Csernansky, J. Morris; P50 AG05681, P01 AG03991, P01 AG026276, R01 AG021910, P20 MH071616, U24 RR021382. Data collection and sharing for the ADNI project was funded by the Alzheimer's Disease Neuroimaging Initiative (ADNI) (National Institutes of Health Grant U01 AG024904) and DOD ADNI (Department of Defense award number W81XWH-12-2-001—adni.loni.usc.edu). As such, the investigators within the ADNI contributed to the design and implementation of ADNI and/or provided data but did not participate in analysis or writing of this report. A complete listing of ADNI investigators can be found at http://adni.loni.usc.edu/wp-content/uploads/how_to_apply/ADNI_Acknowledgement_List.pdf. ADNI is funded by the National Institute on Aging, the National Institute of Biomedical Imaging and Bioengineering, and through generous contributions from the following: AbbVie, Alzheimer's Association; Alzheimer's Drug Discovery Foundation; Araclon Biotech; BioClinica, Inc.; Biogen; Bristol-Myers Squibb Company; CereSpir, Inc.; Cogstate; Eisai Inc.; Elan Pharmaceuticals, Inc.; Eli Lilly and Company; EuroImmun; F. Hoffmann-La Roche Ltd and its affiliated company Genentech, Inc.; Fujirebio; GE Healthcare; IXICO Ltd.; Janssen Alzheimer Immunotherapy Research & Development, LLC.; Johnson & Johnson Pharmaceutical Research & Development LLC.; Lumosity; Lundbeck; Merck & Co., Inc.; Meso Scale Diagnostics, LLC.; NeuroRx Research; Neurotrack Technologies; Novartis Pharmaceuticals Corporation; Pfizer Inc.; Piramal Imaging; Servier; Takeda Pharmaceutical Company; and Transition Therapeutics. The Canadian Institutes of Health Research is providing funds to support ADNI clinical sites in Canada. Private sector contributions are facilitated by the Foundation for the National Institutes of Health (www.fnih.org). The grantee organisation is the Northern California Institute for Research and Education, and the study is coordinated by the Alzheimer's Therapeutic Research Institute at the University of Southern California. ADNI data are disseminated by the Laboratory for NeuroImaging at the University of Southern California.

**Conflicts of Interest:** The authors declare no conflicts of interest. The funders had no role in the design of the study; in the collection, analyses or interpretation of data; in the writing of the manuscript; or in the decision to publish the results.

**Abbreviations**

The following abbreviations are used in this manuscript:

| | |
|---|---|
| AD | Alzheimer's disease |
| ADNI | Alzheimer's Disease Neuroimaging Initiative |
| CI | Confidence interval |
| FCM | Fuzzy c-means |
| FSL-BET | Brain Extraction Tool |
| FSL-SIENA | Structural Image Evaluation, using Normalization, of Atrophy |
| GMM | Gaussian mixture model |
| HM | Histogram matching |
| KDE | Kernel density estimation |
| KL | Kullback–Leibler |
| MNI | Montreal Neurological Institute |
| MP-RAGE | Magnetization Prepared-RApid Gradient Echo |
| MRI | Magnetic resonance imaging |
| NC | Normal control |
| OASIS | Open Access Series of Imaging Studies |
| SD | Standard deviation |
| WS | WhiteStripe |

**Appendix A. Histograms of Intensity before and after Intensity Standardisation**

We measured standardisation quality based on the degree of overlap between histograms of intensities of baseline and follow-up scans using the KL divergence. To further illustrate the effect of intensity standardisation, we plotted the histograms of intensity before and after standardisation. The resulting figures for both OASIS and ADNI can be seen in Figures A1 and A2. In general, standardisation reduced inter-subject intensity differences. However, discrepancies in the left-most tails of the resulting histograms indicate there are remaining unwanted technical variations that intensity standardisation does not cope with. In the case of WhiteStripe, this situation has been also noted in previous works in the field [28].

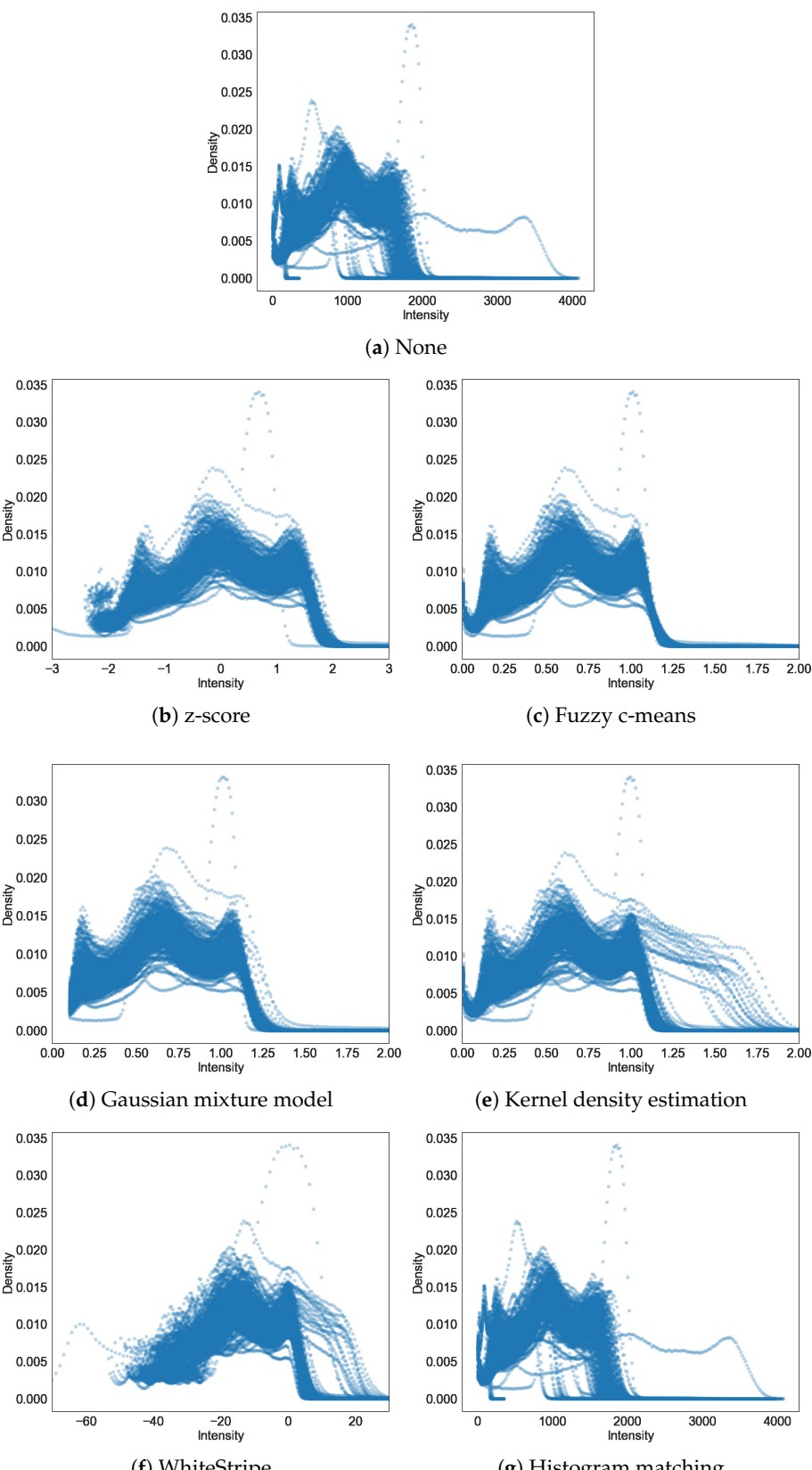

**Figure A1.** Histograms of intensity of OASIS scans before and after intensity standardisation.

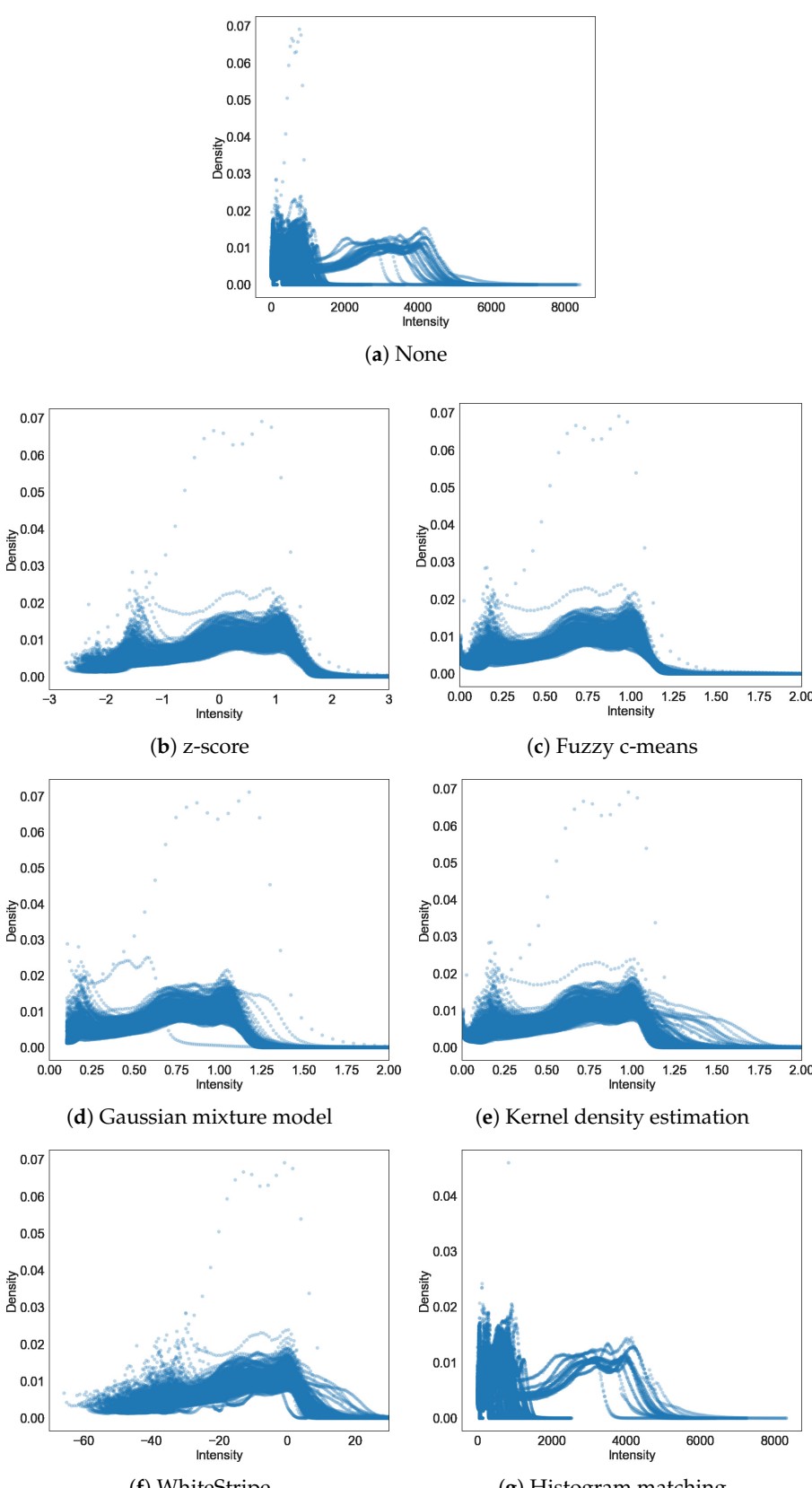

**Figure A2.** Histograms of intensity of ADNI scans before and after intensity standardisation. We noticed that intensity variations depend on acquisition site. In particular, scans from site 002 exhibited higher intensities than any other site.

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
