# Peer review of "Evaluating the Effect of Intensity Standardisation on Longitudinal Whole Brain Atrophy Quantification in Brain Magnetic Resonance Imaging"

_applsci, doi:10.3390/app11041773_

Round 1

Reviewer 1 Report

This paper describes a set of interesting and important experiments studying the value of intensity normalization for the quantification of accuracy. The strengths include the comparison of a variety of intensity normalization methods as well as the use of two datasets and several assessments of the performance of the atrophy estimation. However, there are several areas that require significant attention:

First, on the sample considered:

  1. How were the subsamples selected from OASIS (n=122) / ADNI2 (n=43) ?

  1. What equipment was used for imaging?

  1. In OASIS, what kinds of dementia were included?

  1. How far apart are the scan-rescan in OASIS?

  1. Were longitudinal scans from OASIS used? How far apart were these?

  1. Were the controls matched on covariates? No data on sex / age by diagnosis group were provided.

  1. The distribution of inter-visit times would be helpful.

Secondly, on the intensity normalization methods:

  1. The WhiteStripe method doesn’t require any alignment to MNI; it is generally only helpful for skull-on images. On T1-weighted imaging, it also doesn’t use the largest peak but rather the latest peak (highest T1 intensity after smoothing and truncation of the tail). Indeed, the method has been shown to perform well in the presence of white matter lesions from both multiple sclerosis (as the authors point out) and other aging-related changes (in fact, in the ADNI). The description of the WhiteStripe method in this manuscript seems to be a modified version compared to that which was originally published. This should be discussed.

  1. Upon further inspection of the referenced GitHub, it seems that the authors used a third-party implementation of WhiteStripe. This implementation uses a kernel density smoother as opposed to the penalized spline that was originally employed and validated. Without further validation of the Python implementation compared with the original implementation in R, the results of this study become difficult to interpret. Comparison with the gold-standard implementation is necessary. It is important to note that the RAVEL normalization also depends on this, so re-doing analyses using the R implementations of both of these is required.

  1. A key limitation to the RAVEL framework is the requirement for (nonlinear) registration to a template space. This is absolutely necessary – not something that can be skipped. This should be described and discussed. It is unclear how this affects the subsequent image processing in this study under the various scenarios.

On the comparisons conducted:

Power analysis

  1. Equation (6) is a bit confusing – perhaps replacing the denominator with a single symbol and explaining this would be helpful. Indeed, “the effect size” seems poorly defined – 25% treatment effect would mean a 25% reduction in the effect size, i.e. difference between diseased and control? So if methods produce a larger difference between diseased and control, then the 25% treatment effect needed to achieve would be larger? Not clear why 25% treatment effect was chosen, rather than some other percentage or simply using the diagnosis effect size.  

  2. Does “due to possible outliers in the data” (line 176) refer to the choice of the non parametric Mann-Whitney U-test, or some other aspect of the effect size formulation? (Also the results don’t mention whether or not there were in fact outliers.)

  3. When deriving effect sizes, why not fit a linear model to control for potential differences in covariates between diseased and control?

  1. It is unclear what “we included all percentages of brain volume change…” refers to; are they looking at the difference in atrophy rates between heathy and dementia?

  1. What is the rationale that the method that yields smallest sample size is best? Why should we believe that to be true?

  1. The authors use the term “treatment effect”, which is a bit confusing. Is this the same or different from diagnosis effect?

General considerations concerning comparisons

  1. There seems to be a tension between the 2 evaluations – KL divergence assumes that it is best if within-subject longitudinal scans are more similar to each other, while the power analysis assumes that it is better if the dementia group shows greater change than the control group.

  1. A key question relates to the shape of the distributions (and their scale) after normalization. Perhaps showing histograms or density estimates pre- and post-normalization might shed additional light on the differences in the results.

On the results:

  1. WhiteStripe and z-scoring are very similar – they are both simple linear transformations of the intensities. The very large discrepancies in the results should be examined further.

  1. The RAVEL results in the ADNI seem to be at odds with those reported in the Fortin et al. paper, which showed improved comparability in intensities after RAVEL compared with the raw data. This should be further examined. Are the RAVEL-normalized data from Fortin et al. available for comparison? Or is code available to reproduce these?

  1. Table 2. What is the median (IQR) %? This is the median annual % change?

  1. How is the effect size derived precisely? From Equation 5?

  1. Why is effect size larger for BET –f 0.2 but sample size smaller for BEaST?

  1. Figure 2. What are the numbers along the top of plots? They are not defined in the figure/caption. If these are p-values this should be stated. Ditto for Figure 3.

  1. A sample size reduction of 95% seems rather large. This should be discussed.

  1. How are there asterisk values for RAVEL if FSL-SIENA failed to process scans standardized with RAVEL? Is this just some scans? How many? [Table A1 clarifies that it was 6 scans, but this should be stated in the main paper.] How should one make sense of the large reduction in sample size for this case?

Overall limitations:

A key limitation to the project is the use of only one method for assessing brain volume change. It is quite difficult to interpret what might generalize to other image analysis pipelines.

Author Response

We would like to thank the reviewers for their detailed and constructive appraisals. We have addressed all of the reviewers’ suggestions and criticisms in this revision. We believe this has enhanced our work significantly. Please, find attached our responses.

Reviewer 2 Report

The authors presented an interesting study trying to find a solution for the intensity variance between scans for longitudinal studies to improve the assessment of brain atrophy. This study can help readers to choose a suitable method of intensity normalization. This topic is essential to be investigated to reduce the intensity variance that may be inherent by the MRI scanner itself. However, the presented study focused only on the intensity normalization, without considering some important complimentary methods: tissue inhomogeneity and standardized the nonbrain tissue stripping.

I have some concerns regarding the study:

  1. Abstract: What is the time period between the two scans?
  2. Through the method section, the author did not show any details about the periods between scans and how that influence the results from two prospective, the existence of atrophy and the level of intensity variation from scans to scan over a different period of times, for instance, after one week, one month, one year or more.
  3. There are six methods (sections) describing the intensity normalization. I suggested to separate the clustering into two sections. In the method section, I recommend the authors state that they applied all the methods before and after registration.
  4. Line 69 scan-rescan Does the time is a matter, for example, within a few hours rescan?
  5. What are the MRI sequences that were used in this study?
  6. Table 1: the “Annual visit” is not clear, approximately. What does that mean? What is the exact period?
  7. Line 87: “before and after” It should be clarified that means trying the effect of intensity standardization in two separate experiments, one before and one after.
  8. The equation variables should be defined after each equation, for example, N, M, L, X, … etc.
  9. Line 91: why the absolute ICV used in the equation and does the ICV is the brain’s total volume or the average intensity of the brain that supposed to be used for the normalization. It needs to be clarified.
  10. Section 2.3.2: Why the method depends on the WM for normalizing the intensity and not choosing the GM or CSF. Does this method target the FLAIR images for other applications rather than brain atrophy? Does that may influence the KL results.
  11. What is the registration method used to register the MRI images to the MNI space? Does it linear or non-linear, and does that influence the intensity variation or standardization?
  12. Line 113: “reference” Which one is considered the reference scan baseline or follow-up visit? Does it perform after nonbrain tissue stripping or before?
  13. Line 122: What is the degree of error of the nonbrain tissue stripping method? Does it a massive amount of nonbrain tissue remaining or a small portion of the skull or what? Does the author consider manual editing for correcting these errors?
  14. Section 2.3.5: Does it segmented to get the WM? How does the normal-appearing WM create?
  15. Line 132: “Center of the Head” Is it possible to be the CSF? Or how to specify the Center of the head, see Segonne et al., Neuroimage 22 (3), 1060–1075 and Bahrani et al., Journal of Neuroscience Methods 327 (2019) 108391.
  16. Line 149: Does that mean you need to segment the CSF and normalize it to estimate the weight or randomly picked values? Furthermore, does the image treated without intensity correction method N3-correction or all based on the raw data without N3or4 correction?
  17. Section 3.1: Does it require manual editing. Does the method consider the Center of the head (Center of gravity) to improve the skull stripping? See Segonne et al., 2004 and Bahrani et al., 2019. That explaining the standardization method for skull stripping
  18. What is the FSL stripping option used? Maybe the parameters combination that were chose for BET is not enough, for example, the default is 0.5, and you tried 0.1 and 0.2
  19. Line 220: Do you mean a single session after a few minutes, or what is the period?
  20. Table 3: Why the ADNI result shows a big difference compared to the OASIS cohort?

Author Response

(The authors gave the same response as above.)

Round 2

Reviewer 1 Report

We thank the authors for their thoughtful revision.

Author Response

Reviewer#1, Point # 1: We thank the authors for their thoughtful revision.

Author response: We thank the reviewer for their thoughtful comments and overall effort.

Reviewer 2 Report

The authors were responded properly to the comments. There are minor comments as shown below need to be addressed:

Point #19: It is probably the difference between studies related to the MRI sequence acquisition parameters. I recommend the authors add the acquisition parameters for the T1-weighted image for both studies to show if that caused the differences.

Appendix A needs some explanation or legends.

There is a typo in Fig. 5 (delete one AND).

Remove the ICV from the abbreviations and check if there are more.

Sections 2.3.2 and 2.3.3. I suggest replacing "it" with the "mean."

Section 2.3.2. I suggest replacing "Task at hand" with something that makes it more obvious to the reader.

Section 2.3.3., first line: Do the authors mean "Based on WM."

ADNI section, typo MR-RAGE.

yousecond person pronoun; the person addressedMore (Definitions, Synonyms, Translation)

Author Response

We thank the reviewer for the thoughtful comments, detailed revision and overall effort, which has enhanced significantly the quality of our work. Please, find attached our responses in the attachment.
